

# The OCEAN ICE mooring compilation: a standardised, pan-Antarctic database of ocean hydrography and current time series

Shenjie Zhou[1], Pierre Dutrieux[1], Claudia F. Giulivi[2], Adrian Jenkins[3], Alessandro Silvano[4], Christopher Auckland[1], E. Povl Abrahamsen[1], Michael Meredith[1], Irena Vaňková[5], Keith Nicholls[1], Peter E. D. Davis[1], Svein Østerhus[6], Arnold L. Gordon[2], Christopher J. Zappa[2], Tiago S. Dotto[7], Ted Scambos[8], Kathryn L. Gunn[4], Stephen R. Rintoul[9], Shigeru Aoki[10], Craig Stevens[11], Chengyan Liu[12], Sukyoung Yun[13], Tae-Wan Kim[13], Won Sang Lee[13], Markus Janout[14], Tore Hattermann[15], Julius Lauber[15], Elin Darelius[16], Anna Wåhlin[17], Leo Middleton[17], Pasquale Castagno[18], Giorgio Budillon[19], Karen J. Heywood[20], Jennifer Graham[21], Stephen Dye[21], Daisuke Hirano[22], Una Kim Miller[23]

[1]British Antarctic Survey, Cambridge, UK
[2]Lamont-Doherty Earth Observatory, Columbia University, Palisades, USA
[3]Northumbria University, Newcastle, UK
[4]University of Southampton, Southampton, UK
[5]Los Alamos National Laboratory, Los Alamos, USA
[6]Norwegian Research Centre (NORCE) and Bjerknes Centre for Climate Research, Bergen, Norway
[7]National Oceanography Centre, Southampton, UK
[8]University of Colorado Boulder, Boulder, USA
[9]Commonwealth Scientific and Industrial Research Organisation (CSIRO), and Australian Antarctic Program Partnership, University of Tasmania, Hobart, Tasmania, Australia
[10]Hokkaido University, Sapporo, Hokkaido, Japan
[11]National Institute of Water and Atmospheric Research (NIWA), Wellington, New Zealand
[12]Southern Marine Science and Engineering Guangdong Laboratory (Zhuhai), Zhuhai, China
[13]Korean Polar Research Institute, Incheon, South Korea
[14]Alfred Wegener Institute for Polar and Marine Research, Bremerhaven, Germany
[15]Norwegian Polar Institute, Tromsø, Norway
[16]Geophysical Institute, University of Bergen and the Bjerknes Centre for Climate research, Bergen, Norway
[17]University of Gothenburg, Gothenburg, Sweden
[18]University of Messina, Messina, Italy
[19]Parthenope University of Naples, Naples, Italy
[20]University of East Anglia, Norwich, UK
[21]Centre for Environment, Fisheries and Aquaculture Science, Lowestoft, UK
[22]National Institute of Polar Research, Tokyo, Japan





[23]University of Rhode Island, Kingston, USA
*Correspondence to*: Shenjie Zhou (shezhou@bas.ac.uk)
**Abstract.** Continuous moored time series of temperature, salinity, pressure and current speed and direction are of great
importance for understanding the continental shelf and under-ice-shelf dynamics and thermodynamics that govern water mass
transformations and ice melting in and around Antarctic marginal seas. In these regions, icebergs and sea ice make ship-based
mooring deployment and recovery challenging. Nevertheless, over decades, expeditions around the fringe of Antarctica
sporadically deployed and recovered hundreds of moored instruments, including those facilitated through ice shelves
boreholes. These datasets tend to be archived in a wide range of data centres, with, to our knowledge, no clear format
standardisation. As a result, systematic analysis of historical mooring time series in the marginal seas is often challenging.
Here we present the first version of a standardised pan-Antarctic moored hydrography and current time series compilation,
with broad international contributions from data centres, research institutes and individual data owners. The mooring records
in this compilation span over five decades, from the 1970s to the 2020s, providing an opportunity for a systematic study of the
pan-Antarctic water mass transport and shelf connectivity. As a demonstration of the utility of this compilation, we present
spectral analysis of the compiled current velocity time series, which unsurprisingly shows the dominating presence of tidal
variability within most records. This component of the variability is fitted using multi-linear regression to tidal frequencies,
and the tidal fit is removed from the original time series to leave detided variability. Recalling that records are limited to
months to years in duration, the latter is predominantly composed of synoptic (3-10 days period), intraseasonal (10-80 days)
and seasonal (~6 months-1 year) variability. The spatial distribution of the kinetic energy integrated within each frequency
band (tidal and non-tidal) is presented and discussed within respective regional contexts, and future avenues of research are
proposed. This data compilation is assembled under the endorsement of Ocean-Cryosphere Exchanges in ANtarctica: Impacts
on Climate and the Earth System (OCEAN ICE) project (https://ocean-ice.eu/) funded by the European Commission and UK
Research and Innovation. It is available and regularly updated in NetCDF format with the SEANOE database at
https://doi.org/10.17882/99922 (Zhou et al. 2024a).
**1 Introduction**
The Antarctic continental shelves host multiple sites of unique water mass formation: coastal polynyas are the site of intense
ocean heat loss to the cold polar atmosphere, freezing the sea surface and creating dense salty waters known as High Salinity
Shelf Water (HSSW) via the associated brine rejection. In some parts around the Antarctica, further heat loss via interaction
with the Antarctic ice sheet creates supercooled, saline water masses, known as Ice Shelf Water (ISW), which, together with
HSSW, form the precursors of the most voluminous water mass, Antarctic Bottom Water, that fills global abyssal ocean
(Richardson et al. 2005, Li et al. 2023). Closer to the surface, freshwater resulting from sea ice and ocean-driven glacial melt
of the ice sheet modulates the interactions between atmosphere, ocean and sea ice (Bronsealer et al., 2018, Haumann et al.,
2020). It also modifies exchanges between adjacent seas (e.g., Jacobs et al., 2022) and with the Southern Ocean, as the slope
current and front associated with lateral gradients of temperature and salinity form a dynamic cross-shelf barrier (Thompson



et al., 2018). Through turbulent mixing and modulation of source water properties, shelf sea processes impact the deep ocean
ventilation. These processes influence climate by setting the strength and properties of the overturning circulation and the
exchange of heat and moisture with the atmosphere. Capturing the processes governing the formation and transport of these
water masses is challenging using observations due to logistical difficulties for ships to access these regions readily. As a
result, our knowledge of the freshwater budget over the Antarctic continental shelves and Southern Ocean is poor, with
important repercussions for climate modelling and sea level rise projections (Heywood et al., 2012).
Recent models and observations from a few locations (e.g. Han et al. 2023, 2024) have highlighted how tidal currents and
topographic Rossby waves can promote the mixing and descent of dense water plumes along the continental slope. However,
it remains unclear if these findings can be generalized to all dense water outflows. Similarly, the transport of glacial meltwater
affects oceanic processes downstream, including ice shelf melting, sea ice formation and the creation of dense shelf waters,
especially in the West Antarctic sector from the West Antarctic Peninsula to the Amundsen and Ross Seas (Nakayama et al.,
2020, Jacobs et al., 2022, Dawson et al., 2023, Flexas et al., 2024). However, due to the lack of observations over most of the
Antarctic continental shelf, little is currently known about the connectivity of the circum-antarctic shelf seas.

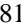



**Figure 1: (a) A comparison between the OCEAN ICE moored time series compilation and Southern Ocean Observing System (SOOS, https://www.soosmap.aq/) map metadata information. The SOOS mooring map information is retrieved from the SOOS map webpage by specifying the instrument type as 'fixed platform' in the interactive webpage. Each square represents a record in either SOOS map or OCEAN ICE compilation. The red squares are those contained in the OCEAN ICE compilation. Hollow squares with orange crosses are valid sites in SOOS map (with status of recovered or deployed). The overlapping sites across the OCEAN ICE compilation and SOOS map are then denoted as a red square with an orange cross on top. Blue crosses are those sites in SOOS map that are shown as either planned, failed or unknown status. These sites are either invalid or the data remains to be released. (b) An example of depth-averaged current vectors and the variance ellipses denoting the cross-stream and along-stream current variabilities for a group of shelf break moorings in the southern Weddell Sea (Darelius et al. 2023). (c) Bottom temperature and (d) bottom salinity over the continental shelf (<1000 m) is shown to distinguish different thermal regimes across the Antarctic**





**continental shelves. Bottom properties are computed as the mean value over the bottom 150m from the seabed from a climatology**
**product constructed from the hydrographic profiles (Zhou et al., 2024b, Zhou et al. in prep). Bathymetry, in grey shading, is from**
**RTopo 2.0.4 (Schaffer et al., 2016).**
Here, we present the first version of a standardized compilation of historical moored time series that have been deployed over
the past 50 years on the Antarctic continental shelves and slopes and present a rapid overview of energetic characteristics of
ocean circulation on and off the continental shelves along with their hydrographic context. This compilation includes
temperature, salinity, pressure and ocean current time series south of 60°S (**Fig. 1a**). The time series are freely and publicly
accessible in a standardized format at SEANOE (https://www.seanoe.org/data/00887/99922/, Zhou et al. 2024a). We intend to
maintain and enhance the compilation on an annual basis and welcome contributions for inclusion in future releases. This
mooring compilation is endorsed by the OCEAN ICE project funded by the European Commission and UK Research and
Innovation, and we refer to this data compilation as the OCEAN ICE mooring compilation herein. This dataset aims to provide
opportunities for regional or systematic pan-Antarctic studies on water mass transport, formation processes and shelf
connectivity.

## 2 Data sets and processing

### 2.1 Overview

In the OCEAN ICE mooring compilation, we collected 521 mooring time series, covering 470 deployment sites (**Fig. 1a**). The
comparison with the Southern Ocean Observing System (SOOS, https://www.soosmap.aq/) mooring map which aims at
compiling links to datasets from all past endeavours shows that our compilation includes additional  mooring records, e.g. in
front of the Ross Ice Shelf and from instruments deployed through boreholes in the Ross and Amery ice shelves. Additionally,
whilst SOOS map provides an overview of moorings location and metadata in their interactive web map, the actual datasets or
links directly leading to the datasets are not provided. Therefore, the OCEAN ICE mooring compilation is an effort to improve
spatial coverage and directly provide publicly available datasets., calling on experienced international collaborators to obtain
a compilation that is as complete as possible. This effort will be continued, and the compilation will be expanded in future
annual releases. At this stage, we have collated data from regions and features not represented in SOOS, including the Antarctic
Slope Current and Antarctic Bottom Water transport over the slope current from places such as the southeastern Weddell Sea
(Graham et al. 2013), Princess Elizabeth Trough (Heywood et al. 1999) and Australian-Antarctic Basin (Peña-Molino et al.,
2016), in addition to those that have been logged in the SOOS map. The mooring time series are acquired from various sources.
Some of them are archived in public databases such as Pangaea Data repository, British Oceanographic Data Centre, UK Polar
Data Centre, US Antarctic Program Data Center, Australian Antarctic Data Centre, Korea Polar Data Centre, Norwegian
Marine Data Centre and Norwegian Polar Data Centre. Others are stored in places that are less commonly considered as
Antarctic mooring data centres such as NCEI/NOAA, or local databases hosted by individual institutes such as Lamont-
Doherty Earth Observatory of Columbia University and Oregon State University (e.g. the OSU Buoy Group,
https://cmrecords.net/history.html). A list of mooring record source links is stored with our database and is available as an
additional file on SEANOE where the OCEAN ICE mooring compilation is published. **Fig. 1b** showcases a regional example



of current metre measurements and some basic information, namely the depth-averaged current vectors along with variance
ellipses depicting the along-stream and cross-stream velocity variabilities. The broad spatial spread of these mooring sites
covers different thermohaline regimes over the continental shelves as indicated by the climatological bottom water
temperature/salinity in **Fig. 1c** and **Fig. 1d**, from the colder and saltier dense water formation site of the Ross, Weddell, and
Cosmonaut seas to the warmer Amundsen and Bellingshausen seas. One notable characteristic of this dataset is its typical mid-
water column to near-seabed sampling bias. Indeed, most mooring deployed in Antarctic shelf seas tend to avoid sampling the
near surface region where drifting icebergs can damage instrumentation.

### 134 2.2 Data standardisation

All the mooring time series are standardized and re-formatted into individual NetCDF files for each mooring site, following a
consistent file structure. The source of individual datasets is also provided, allowing further investigations and analysis of the
processing steps applied to each time series before we obtain them. We are not always aware, for example, if corrections for
current-induced motions of the sensors or the magnetic declination corrections on current direction have been applied for each
individual mooring. For moorings containing multiple instruments, to acknowledge the fact that these instruments are sampled
at different frequencies and over different periods of time, each instrument is accompanied by its own time vector in the
NetCDF file. **Table 1** shows examples of variable lists from three types of the most commonly deployed instruments -
Temperature, Conductivity and Pressure logger (e.g., SBE37 MicroCAT), Acoustic Doppler Current Profiler (ADCP, e.g.,
Teledyne RDI 75kHz ADCP), and current meter (e.g., Aanderaa Rotor Current Meter). Note that in the final form of the
mooring file, we retain the original sampling frequency for all the mooring instruments as we received it (some were already
processed and averaged), avoiding modifications of the temporal resolution as much as possible, to ensure broader use of these
mooring records for analysing processes spanning sub-daily to interannual time scale ranges. Additionally, we performed
minimum data clean up, solely replacing bad data identified by various flags or unrealistically large numbers with NaNs. No
additional interpolation/extrapolation is applied to retain the original mooring time series.

| Filename/instrument name | Variable names | Attributions and units |
|---|---|---|
| SB_2013.nc/SBE37 MicroCAT | Instrument_01_info | Instrument type = sbe37_7224 |
| | Instrument_01_date | Days since 1950-01-01 00:00:00 |
| | Instrument_01_depth | Instrument depth (m) |
| | Instrument_01_press | Instrument water pressure (dbar) |
| | Instrument_01_salin | In-situ salinity (PSU) |
| | Instrument_01_temp | In-situ temperature (°C) |
| SB_2013.nc/Teledyne RDI 75kHz ADCP | Instrument_01_info | Instrument type = rdi_adcp_75khz_18447 |
| | Instrument_01_date | Days since 1950-01-01 00:00:00 |
| | Instrument_01_bindepth | ADCP depth bins (m) |
| | Instrument_01_binpress | ADCP pressure bins (dbar) |
| | Instrument_01_u | Current velocity zonal component (cm/s) |
| | Instrument_01_v | Current velocity meridional component (cm/s) |
| FDRAKE75_M12.nc/Aanderaa Rotor Current Meter | Instrument_05_info | Instrument type = Aanderaa_RCM5 |
| | Instrument_05_date | Days since 1950-01-01 00:00:00 |
| | Instrument_05_depth | Instrument depth (m) |





| Instrument_05_press | Instrument water pressure (dbar) |
| Instrument_05_u | Current velocity zonal component (cm/s) |
| Instrument_05_v | Current velocity meridional component (cm/s) |

**Table 1. An example of the variable lists for a SBE37 MicroCAT, a Teledyne RDI 75kHz ADCP equipped on mooring SB_2013.nc,**
**and an Aanderaa. Rotor Current Metre equipped on mooring FDRAKE75_M12.**
Individual instrument information is available in the variable descriptions, including the name of the instrument and its serial
number, if available. For those datasets where the instrument serial numbers were not made available to us, instruments with
identical names are differentiated by labelling them numerically in order of the mounted depth along the mooring line. Both
acoustic Doppler current metre and current profiler (ADCP) records are included in this compilation, with units of cm/s, at
time _date (days since January 1st, 1950) and depth _depth (m). Specifically, ADCP data are stored in two-dimensional M×N
arrays, M being the number of records, N being the number of vertical bins, and in this case, depth is also two-dimensional.
All temperature (°C, IPTS-90) and salinity (Practical Salinity Unit) data stored are in-situ measurements, again recorded at
depth _depth (m) and time _date (days since January 1st, 1950).

**2.3 Inferred Tidal Energy and Eddy Kinetic Energy**
In the following, we present an initial estimate of the frequency content and spatial distribution of the kinetic energy contained
within the compiled records. We isolate the tidal components of the variability from individual records using a multi-linear
least square fit to tidal components (UTide, Codiga 2011), providing us with the fitted tidal harmonics. By removing the fitted
tidal harmonics from the original record, 'detided' time series can be used to reflect the energy associated with non-tidal
processes, hereafter referred to as eddy kinetic energy (EKE).

**Figure 2** shows the probabilistic distribution function (PDF) plot of the current speed spectra integrating all the mooring sites
where current metres or ADCPs were deployed. For this exercise, ADCP records are first averaged in the vertical over all
recorded bins, giving a single time series per ADCP, from which the spectra are then extracted to be more readily compared
with single point current meters. Figure 2a shows the spectra PDF resulting from the original time series. The heat map pattern
is predominantly characterised by a classic red spectrum, with pronounced tidal energy peaks at the semi-diurnal (0.48 to 0.57
days), diurnal (0.9 to 1.2 days) and fortnightly (13.7 to 14.8 days) frequency bands. Distinct peaks are also visible for higher
and lower frequency tidal harmonics. The spectra PDF of the fitted tidal harmonics is shown in **Fig. 2b** and highlights the
presence of various tidal harmonics and their elevated energy levels. The tide-free or detided spectra PDF (**Fig. 2c**) show a
smoother red form, with an overlay of relatively elevated energy peaks around the semi-diurnal frequency range and a smaller,
more diffuse energy bump around the synoptic timescales with periods contained between 3 and 10 days.

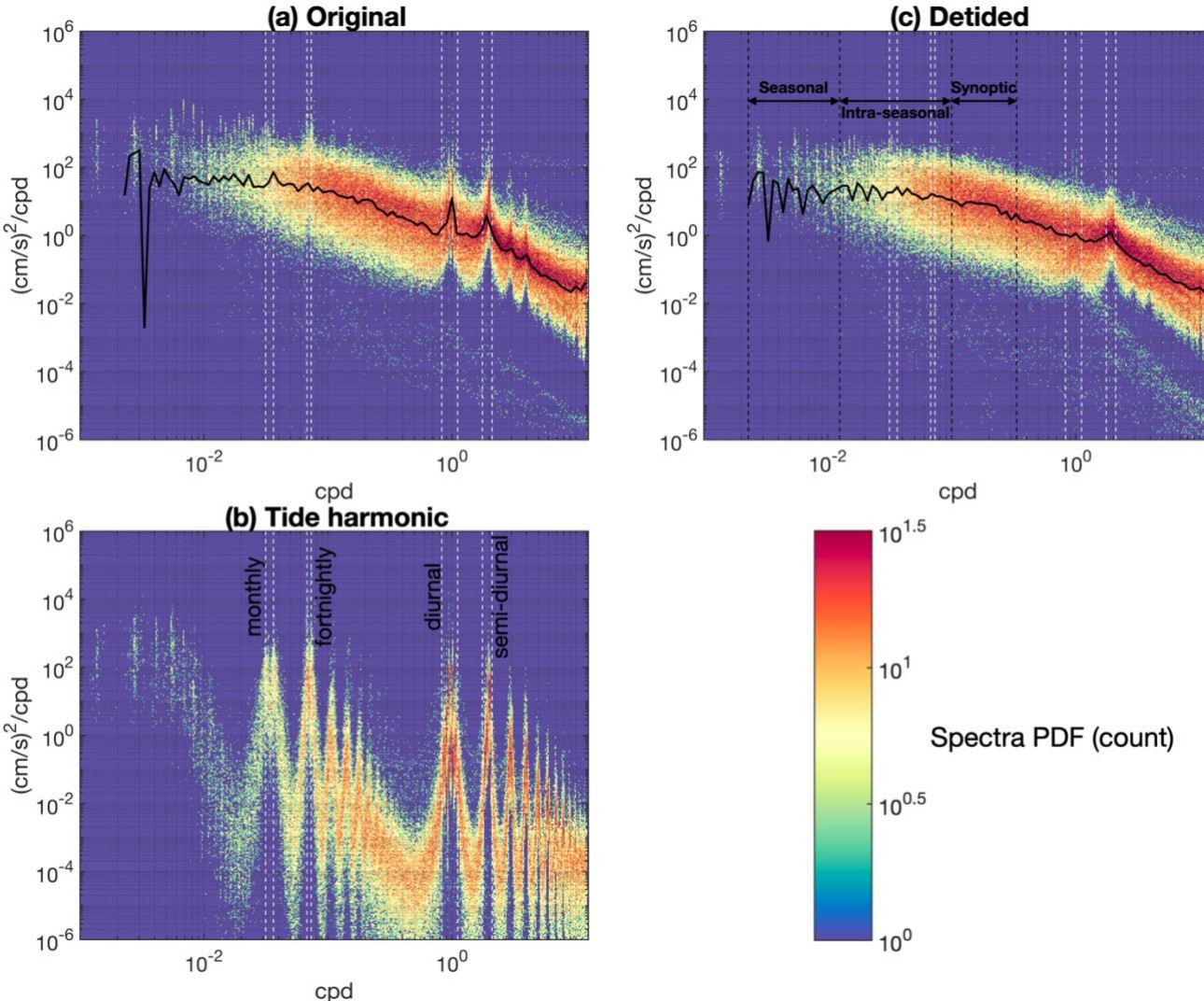

**Figure 2.** The colour shading shows the spectra probabilistic distribution function (PDF) of all the (a) original velocity time series, (b) the tidal harmonics and (c) detided velocity time series. Dashed vertical lines denote the upper and lower frequency bounds for monthly (27.6 to 31.8 days), fortnightly (13.7 to 14.8 days), diurnal (0.9 to 1.2 days) and semi-diurnal (0.48 to 0.57 days) tidal components. Higher/lower values (red/blue) mean that the level of energy is more frequently observed at given frequency across different mooring time series. Black dashed lines in panel (c) show the upper and lower frequency bounds for synoptic (3-10 days), intraseasonal (10-80 days) and seasonal (80 days-1.2 years) timescales. Black lines in panel (a) and (c) denote the power density level that is the most frequently counted (i.e. the mode) at each frequency range.

To provide a view of the spatial distribution of kinetic energy, we further integrate the spectra of all three sets of time series (original, tidal harmonics and detided) for each record over a set of frequency ranges, representing the semi-diurnal (0.48 to 0.57 days), diurnal (0.9 to 1.2 days), fortnightly (13.7 to 14.8 days), synoptic (3 to 10 days), intraseasonal (10-80 days) and seasonal (80 days to 1.2 years) timescales. An example of the spatial distribution of the kinetic energy before and after the



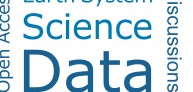

tide removal is shown for the diurnal tidal range in **Fig. 3**. The original time series (**Fig. 3a**) in fact contains a range of
kinetic energy peaks close to the diurnal periodicity, most of which clearly correspond to the exact tidal harmonics (**Fig. 3c**),
such that the detided time series show much lower diurnal peaks. Broadly speaking, most of the ocean kinetic energy at
diurnal (**Fig. 3 b-c**), semi-diurnal (**Fig. 4 a-b**), and fortnightly (**Fig. 4c-d**) periodicity are indeed driven by the tide, and the
detided energy on these three frequency bands is much lower than the original and tidal harmonics estimations (see also **Fig.**
**2**).





**Figure 3. The spatial distribution of the kinetic energy integrated over the diurnal frequency band (0.9 to 1.2 days) using (a) original time series, (b) detided time series and (c) fitted tidal harmonics. Grey shading shows the bathymetry, coloured circles showing the magnitude of the kinetic energy integrated over the diurnal frequency band.**

However, we note that some of the detided records still retain relatively strong energy levels within the diurnal (**Fig. 3b**) and semi-diurnal (**Fig. 4a**) ranges. This property is particularly pronounced in regions where dense shelf water flows out of the Ross Sea, the Cosmonaut Sea (off the Amery Ice Shelf) and the Terre Adélie Sea. The ice front of the Ross and Filchner-Ronne Ice Shelf and the entrance of the Filchner Trough also show elevated levels of diurnal and/or semi-diurnal variability in detided time series. The presence of the semi-diurnal to diurnal variability within the detided records may result from other sources of variability, e.g. overlapping with the inertial range which is closer to semi-diurnal in polar regions, and/or dispersion of tidal energy around the exact tidal harmonic frequency via mixing processes or spectral diffusion in more poorly sampled records. The fact that there is little EKE in the detided records at the daily and fortnightly periodicity may indicate a generally low level of spectral diffusion, though this factor is frequency dependent. It also lends support to a hypothesized importance of inertial dynamics as a source of EKE around the semi-diurnal frequency. We note that high kinetic energy (**Fig. 4b**) is found in mooring sites for detided time series where the current metres were instrumented close to the seabed, where the current is arguably more susceptible to mixing driven by local topography. However, the more detailed analysis required to elucidate the reason for the elevated energy level remaining around tidal frequency ranges within detided time series is beyond the scope of this publication.

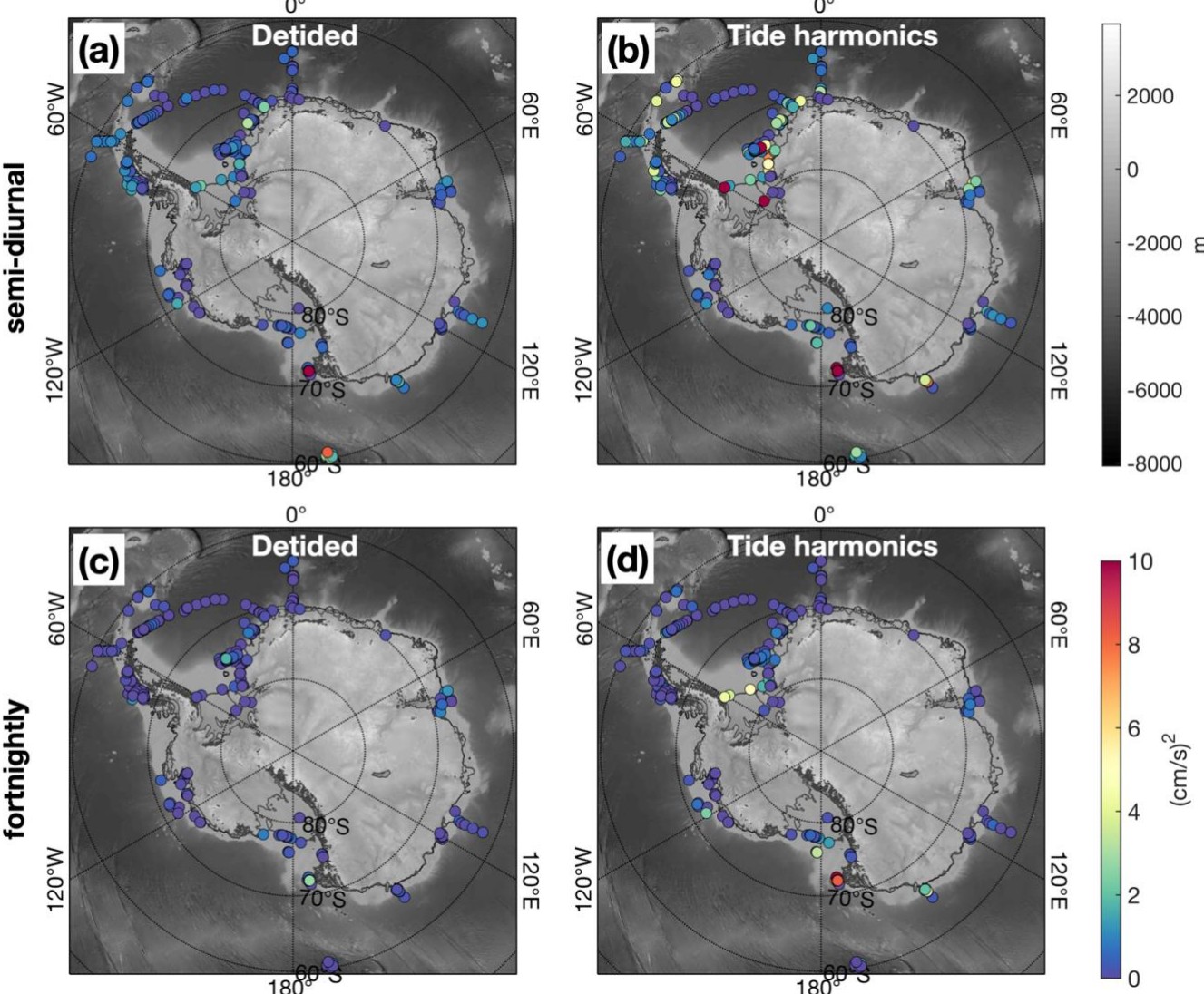

**Figure 4. The spatial distribution of (a) detided kinetic energy integrated over semi-diurnal periodicity (0.48 to 0.57 days), (b) kinetic energy estimated from fitted tide harmonics over semi-diurnal periodicity. (c) Same as a but for fortnightly periodicity (13.7 to 14.8 days). (d) Same as d but for fortnightly periodicity. Grey shading shows the bathymetry, coloured circles showing the magnitude of the kinetic energy integrated over the respective frequency bands.**

At lower frequency, EKE can be divided into three bands (**Fig. 5**), namely synoptic (3 to 10 days), intraseasonal (10 to 80 days) and seasonal (80 to 1.2 years) timescales. The synoptic band shows elevated energy levels along most of the Antarctic continental shelf break (**Fig. 5a**). Inshore, and along glacier fronts, a notable energy distribution pattern emerges - regions corresponding to relatively high depth integrated ocean heat content and associated glacial melt such as the Amundsen and Bellingshausen seas and the Totten and Denman glacier fronts all show relatively low synoptic EKE level, in contrast to the higher EKE levels in regions characterised by cold regimes in front of the Ross, Filchner-Ronne and Amery ice shelves. We

speculate that this difference is associated with the heightened sensitivity of cold regions to synoptic atmospheric variability
through the response of the ocean surface to sea ice formation and stronger katabatic wind events, whilst warmer regions
tend to be more stratified, somewhat insulating the lower part of the water column from surface synoptic variability. In
addition, the dense water produced through the sea ice formation can also lead to strong pulses of outflow and imprints on
the synoptic timescales, leading to heightened EKE levels in cold regimes.






**Figure 5. The spatial distribution of the detided kinetic energy integrated over (a) synoptic (3 to 10 days), (b) intraseasonal (10 to 80 days), (c) seasonal (80 days to 1.2 years) frequency bands. Grey shading shows the bathymetry, coloured circles showing the magnitude of the kinetic energy integrated over the respective frequency bands.**

Another source of EKE can be short coastal waves excited by the atmospheric forcing, dense outflows (Jensen et al., 2013) or resulting from local flow instability (Chavanne et al., 2010). This source of variability, which appears visible in a few bottom pressure records and sea surface height (McKee and Martinson 2020) would probably apply to a broader range of frequencies, from synoptic to seasonal. Interestingly, the regional pattern of spatial variability in EKE revealed above for synoptic time scales also holds for longer (intraseasonal) time scales (**Fig. 5b**). A deeper analysis cross correlating atmospheric and ocean variability and applying coastal wave model responses to wind forcing may help to elucidate the processes driving the synoptic-to-intraseasonal EKE. And a more detailed attribution study investigating the processes associated with sea ice formation, dense water production/outflow and the response of ocean currents to the mechanical stirring from the surface stresses is warranted.

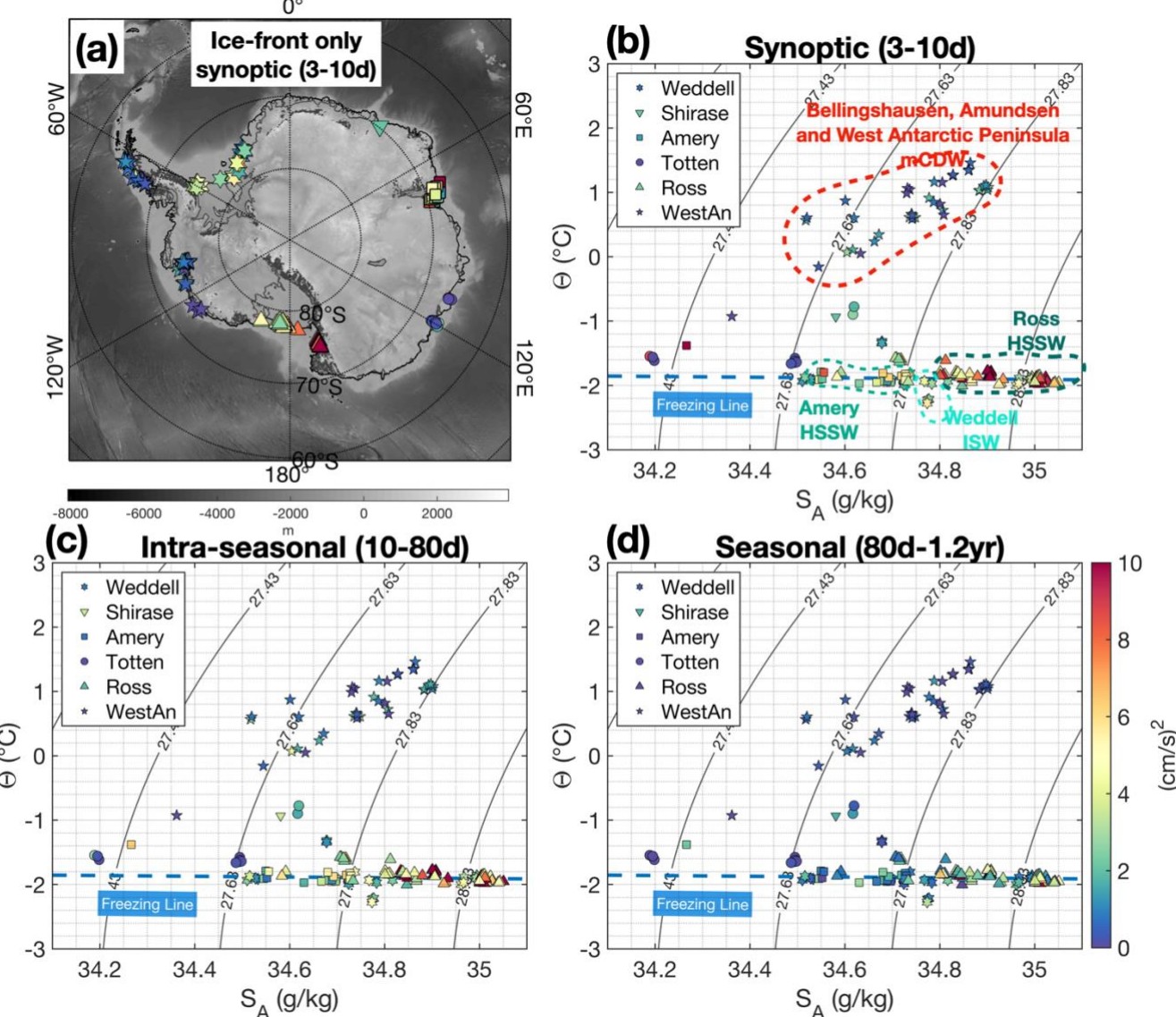

**Figure 6. (a) The locations of ice-front moorings coloured by detided synoptic scale energy. To avoid overlap, data points from the same mooring sites, but with different sampling depths or time periods, were offset by on average 0.5° in longitude and latitude. Grey shading shows the bathymetry. Panel (b)-(d) shows the EKE magnitude integrated over synoptic, intraseasonal and seasonal scales in Θ-S$_A$-space.**

Finally, we present the EKE within the seasonal range (**Fig. 5c**). While the overall EKE level on seasonal timescales is lower than that on synoptic and intraseasonal timescales, the same spatial distribution pattern - whereby EKE levels are enhanced in cold regime shelf seas - still holds. This further highlights the influence of the sea ice thermodynamic and dynamic interactions with the ocean, either through sea ice production and associated deep convection in cold regime shelf seas, and/or modification of the surface stress transmitted by the wind to the ocean via the sea-ice and the downward propagation



of this energy to the lower part of the water column (more frequently sampled in this database). A deeper analysis is needed
to disentangle the roles of each process in each sector, and as a function of time. To further emphasize the regional
correlation between local shelf properties and EKE, we sub-selected ice-front moorings within 50 kilometres of the Antarctic
ice sheet (**Fig. 6a**) and plotted EKE at these locations as a function of climatological bottom temperature and salinity (Zhou
et al. in prep), with colour-coded EKE levels displayed in T-S diagrams (**Fig. 6 b-d**). Data points with elevated EKE over
synoptic to seasonal scales concentrate near and below the freezing line, confirming that elevated EKE occurs over regions
where HSSW are formed (e.g., Filchner-Ronne, Amery and Ross). In contrast, warm ice shelves in West Antarctica
(Amundsen, Bellingshausen and the West Antarctic Peninsula sectors) are mostly quiescent on synoptic-to-seasonal
timescales after all tidal motions are removed.

## 3 Future work

We have highlighted a few potential avenues for future research above. One additional obvious future element of work
remaining at the pan-Antarctic scale is a comparison between observed current tidal harmonics with those predicted by
models. Around Antarctica, the most used tide prediction model is the Circum-Antarctic Tide Simulations (CATS, Padman
et al., 2002, Padman et al., 2008). This tidal model was recently updated (CATS2008_v2023) with improved representation
of coastline, ice shelf grounding line, bathymetry and ice draft (therefore water depth) using the BedMachine Antarctic v3
bathymetry product (Morlighem et a. 2020).  It also incorporates an ice flexure model to reflect the tidal deflection near the
grounding zone (Howard et al., 2024). Below, we present a brief comparison, focusing solely on the magnitude of the K1
tidal component, which aims at prompting further analysis elsewhere.









**Figure 7. (a) Magnitude of the K1 component of the tidal current (cm/s) as predicted in CATS2008 (Padman et al., 2002, coloured background) and that fitted with the OCEAN ICE mooring compilation using UTide (overlaid coloured circles). (b) The difference between OCEAN ICE mooring compilation and CATS2008_v2023 model predictions at each mooring location. Red/Blue means moored observations show larger/weaker K1 magnitude compared with the model. Grey shading shows the bathymetry (c) Scatter plot of OCEAN ICE mooring compilation K1 current magnitude against model prediction. The linear fitting suggests an overall good agreement between two methods, but with significant regional spread. 95% confidence levels are shown as error bars for the observed K1 magnitude estimated using UTide. Colours showing the same difference in K1 current magnitude as in panel (b).**

A broad agreement between observations and predictions is found over the open ocean, off the continental shelves, where the tidal signals are generally weaker (**Fig. 7a-b**). Estimates from two methods tend to drift away more significantly over some of the shelf break mooring sites, potentially because of resonant shelf waves at diurnal frequencies (e.g., Semper and Darelius, 2017). Overall, the tidal information extracted from the OCEAN ICE mooring compilation is generally consistent with the CATS2008_v2023 model prediction in K1 periodicity - the model slightly underestimates the K1 current magnitude compared with the observations suggested by the slope of the linear fitting (**Fig. 7c**). Even with improved BedMachine Antarctica v3 bathymetry information, which does not necessarily provide an accurate bathymetry and ice draft geometry information (Charrassin et al., 2025), errors in the model predictions can still be sourced from a variety of factors, including incomplete representation of the seabed and ice base geometries, leading to inaccurate water-column thickness. These factors are expected to be specifically significant underneath ice shelves and over the continental shelves where sea ice historically precluded detailed observations of the geometry (Padman et al., 2002). The assumption of the barotropicity in tidal currents by CATS2008_v2023 tide model may also become problematic in regions featuring complex topographies or in regions with distinct vertical stratification, where the tidal amplitude and phase has been shown to display variations in depth (e.g., Makinson 2002, Makinson et al., 2006). A more detailed analysis, comparing predictions to observations at other frequencies, or scrutinising the baroclinicity of tidal flows at available observation sites, would help refine the prediction model. Given the importance of tides for numerous processes in Antarctic shelf seas, from ice flexure and impacts on grounding zone positions (e.g. Wallis et al., 2024), migration (Rignot et al., 2024), and ice-ocean interactions (Gadi et al., 2023), improving predictions would be a very useful endeavour. We hope the OCEAN ICE compilation presented here will provide a continuously growing backbone for regional and circum-Antarctic analyses in years to come.

**4 Data Availability**

The OCEAN ICE mooring compilation is published on SEANOE Data Repository with the doi link, https://doi.org/10.17882/99922 (Zhou et al. 2024a). The data publication contains two files: a compressed file (2.8 GB) including all the mooring time series files in NetCDF format and a spreadsheet containing the mooring file names, locations, starting date, end date and the doi link to the original individual data file.

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

**Author contribution**

S.Z. and P.D. jointly conceived the study. S.Z. and P.D. discussed about the needed analysis and figures to present in this work. S.Z. performed all the analysis and figure production. S.Z., P.D. and C.F.G. curated the dataset. S.Z. wrote the original draft. All the authors contributed to the data collating and draft revising.

**Competing interests**

The authors declare that they have no conflict of interest.

**Acknowledgement**

The authors would like to thank all the scientists, project principal investigators, technicians, and ship crew members who are involved in designing, deploying, recovering, and re-deploying all the moored instruments. This moored time series compilation would not have been possible without their work in *in-situ* data acquisition, processing, and data archiving. S.Z. and P.D were supported by OCEAN ICE, which is co-funded by the European Union, Horizon Europe Funding Programme for research and innovation under grant agreement Nr. 101060452 and by UK Research and Innovation. C.F.G was funded by NASA project 19-MAP19-0011 Assessing the Impact of Glacial Melt on the Coupled Climate (grant # 80NSSC20K1158). T.-W. K. was supported by the Korea Polar Research Institute (KOPRI) grant funded by the Ministry of Oceans and Fisheries (grant no. PE25110)