# Peer review of "The OCEAN ICE mooring compilation: a standardised, pan-Antarctic database of ocean hydrography and current time series"

_Earth System Science Data, 2025_

## Referee Comment (RC2)

**Review: "The OCEAN ICE mooring compilation: a standardised, pan-Antarctic database of ocean hydrography and current time series"**

Authors: Zhou et al.

**Summary**

This study presents the first standardized, pan-Antarctic compilation of moored hydrography and current time series, developed through a systematic analysis of historical mooring data from the marginal seas and contributions from international data centers, research institutes, and individual data providers. Spanning over five decades (1970s–2020s), the compilation enables detailed analysis of water mass transport and shelf connectivity across the Antarctic margin. The authors demonstrate the utility of the compilation through spectral analysis, removing dominant tidal signals via multi-linear regression. The detided records, though limited in duration, capture synoptic to seasonal variability, with regional patterns of kinetic energy offering insights for future study.

This dataset is a timely and valuable resource for research along the Antarctic margin, addressing an urgent need for sustained observations in the Southern Ocean. I appreciate the international effort by the group of observational scientists in assembling and standardizing this compilation. I have one major comment regarding Technical Quality, along with several minor comments detailed below. Overall, the manuscript is well written and provides a clear and well-structured contribution to the oceanographic community.

**Technical Quality**

While the compilation brings together an impressive range of historical mooring records, it remains unclear how the authors address uncertainty and error analysis across the dataset. The manuscript notes that only minimal data cleaning was performed, with bad data identified by flags or unrealistic values replaced by NaNs, and no further interpolation or extrapolation applied. However, individual mooring datasets often include important quality control information, such as standard errors, instrument uncertainties, or confidence flags. A more detailed explanation of whether and how such uncertainty metrics were retained, harmonized, or reassessed would enhance the transparency of the compilation and support its appropriate scientific use, particularly given the analyses of detided variability presented in the study.

**Minor comments**

– Figure 2: What causes the sharp drop in the black line at low frequencies in Figure 2a? Since this feature is not present in the detided spectra (Figure 2c), it may be related to tidal energy. A brief explanation would help clarify its origin.

– Given the importance of the detided analysis, will the detided time series be made publicly available alongside the original data? It would also be helpful to clarify the sensitivity of the UTide method to record length, particularly in the case of shorter mooring records.

– Lines 189 & 223: The authors define the seasonal band as spanning 80 days to 1.2 years. Would it be more appropriate to refer to this range as "seasonal-to-annual" to better reflect the upper bound?

– I reviewed the dataset and the provided spreadsheet, which includes useful information such as mooring file names, locations, time ranges, and DOI links to the original data sources. To improve usability, I suggest adding a summary document with more detailed metadata, including instrument type, depth ranges, variable definitions and units, quality control flags, and processing history for each record. This additional information would significantly enhance the dataset's clarity and completeness. I also recommend sharing the analysis code used in the study to further support understanding and facilitate broader use of the dataset.

---

## Author Comment (AC1)

This document contains the final response to comments from both reviewers on the original manuscript entitled "The OCEAN ICE mooring compilation: a standardised, pan-Antarctic database of ocean hydrography and current time series".

Reviewer's comments are in **blue**, and our point-by-point responses are in **black**.

**Reviewer #1:**

**General Comment.**

In this paper Zhou et al. present a large dataset compiling available moored observations of temperature, salinity and current velocities around Antarctica since the 1970s. The dataset is impressive, including close to 500 different datasets, covering a range of key ocean environments across all longitudes around Antarctica. Given the crucial role of Antarctic Ocean and ice processes on climate, extensive re-use of the dataset by the research is warranted and may enable substantial advances in the field. I am sure the community will be very grateful to the authors for their great efforts to put the dataset together. Therefore, I strongly endorse publication of the dataset and associated manuscript in ESSD. However, I have identified a series of potential issues and areas for improvement in the current version of the manuscript and dataset. It would be great if the authors could address my comments and suggestions, or at least respond to them, before I accept the article for final publication.

We thank reviewer's general comments on our manuscript. Our response to reviewer's comments below is in black.

**Specific Comments**

My main concern about the dataset is that it is not very clear how the user would know how much trust they could place on each individual instrument, dataset. Where should the user refer to get information about if and how instruments (e.g. conductivity cells) were calibrated, how was the data treated between acquisition and publication, and about general data quality and flags (and/or whether instrument failure or drift are flagged). I also understand this is a complex task, and it may not be realistic to recover detailed data quality metadata from all deployments, but it would be great if the authors could talk about this a bit more in the manuscript, acknowledge this (important) limitation and say that users should refer to the original datasets, but also give an indication on whether the original files contain some more information.

We agree that this information is not explicitly presented in the published dataset. It is indeed a technically challenging task to harmonise the level of quality control of all the datasets that were acquired with different level of processing. The quality of mooring timeseries has always been lack of consistency. But hopefully our dataset provides a standardised product to some degree. The source link provided in both the metadata spreadsheet and each NetCDF file can lead users to the original source of the data or the data owner to better consult for the processing level of these data. Most of the original dataset that we acquired do not contain specific notes on the processing procedures that have been applied. We did

On a different note, it is great that the authors put together such a massive dataset. However, as I was reading the paper and having a look at the data, I have developed the feeling that the dataset can be a bit daunting, from the users' perspective, due to its richness and diversity. I would suggest that the authors do some effort to further digest the dataset to give a clearer overview of it and make it more accessible. Some suggestions I could come up with are:

 In the manuscript, present one or multiple figures and/or tables with general statistics on the dataset: e.g. periods and seasons covered, record lengths, regions sampled, depths sampled, how many records with TS, velocities, both, etc. One good thing to report would be some statistics on the instrument vertical location with respect to local bottom depth / surface, since you mention there is a "bias" toward deep measurements

We thank reviewer's thoughts on adding more general statistics to better overview the dataset. We included a new Figure 2 in the revised manuscript to illustrate the observing period in length and seasonal coverage, the fraction of each combination of variables measured by each mooring deployment, and averaged instruments depth (normalized) to demonstrate our point of 'bottombiased' instrument installation. The exception for the shallower instruments is two under-ice turbulence clusters measurements in George XI ice shelf and Larsen C ice shelf, both deployed by BAS ice team. The instruments on these two moorings are deliberately mounted close to the ice base and within the ice to measure the ice temperature changes and ice-ocean boundary layer dynamics. The new Figure 2 in the revised manuscript is also attached below.

 In the datafiles. For each of the deployment files, you could add an overview in the metadata about the mooring location (not only latitude and longitude, but name of region), types of instruments and sampling depths and period. A link to the original dataset within files themselves could be useful. In general, richer deployment-specific metadata within each netcdf would be great. I am aware that modifying the whole dataset at this point could be a massive task, so please take this only as a suggestion, I leave to the authors' judgement whether this is a sensible thing to do.

We really appreciate reviewer's comments on including more metadata information. Although we believe that the naming of each mooring regions are always lack of consensus – in some cases the naming of the region can be very specific and in other cases it would have to be represented by the name of the broader area due to lack of naming system in mooring locations. We found that the longitude and latitude is the most consistent way of marking the mooring location.

• I wonder if the authors could share the code used to produce the figures for the preliminary analysis presented in the manuscript. That would give an example to users on how to bulk-access the data and generate some interesting insights from the ensemble of observations.

We believe that the code we used in producing figures and reading files are very basic MATLAB coding and should be easily replaced by other programming languages and can perhaps even be done better and more smartly by other more proficient MATLAB users. However, there is one fundamental piece of code we used for analysing tidal harmonics that needs a bit of instruction – the UTide toolbox. The UTide toolbox and its full instruction are available at https://uk.mathworks.com/matlabcentral/fileexchange/46523-utide-unifiedtidal-analysis-and-prediction-functions.

**In line comments**

Line 98. Maybe provide an early indication of the size of the dataset: e.g. "This compilation includes 521 mooring time series [...]"

We thank reviewer's suggestion, but we feel that the mentioning of data quantity at the beginning of the following section makes a more coherent narrative overall. Considering it is not too far earlier in the text, we decided to keep the original way of mentioning the data quantity.

**Line 109. Could you include some general description of the SOOS mooring dataset and how many more records you are including in your database.**

We found that this point is always a tricky one to phrase, but in short, SOOS map does not provide any links to the actual dataset (in data archiving world, this is called landing page), in other words, there are no landing pages behind all the moorings appearing on SOOS mooring map other than some very brief information about the mooring (i.e., name, country, deployment status). In this database, we have included all mooring data that we can gathered from colleagues in the author list, excluding those are still in the water or being analysed/QC'ed by individuals and groups. Therefore, in terms of 'how many more', it would be 521 timeseries (OCEAN ICE) versus 0 timeseries (SOOSmap). We rephrased the text in the revised manuscript to make this point clearer.

Line 114. There is a typo here ",."

Corrected in the revised manuscript.

Figures 3-5. I felt the scatterplots may look better on log-colour scale, to highlight overall patterns rather than some particularly high values in some locations? Not sure... Also, I suggest annotating the maps with key locations mentioned in the text, e.g. on line 227 and others.

We thank reviewer's comments on colour scale. We agree that the logarithmic colour scale can highlight more detailed regional distribution, but these elevated small differences do not change the overall broad picture of the contrasting in non-tidal motion condition between cold and warm shelves, and the fact that tidal motion dominates the semi-diurnal, diurnal and fortnightly periodicities. Therefore, we decided to keep the linear colour scale in Figure 3-5, but we annotated all the first panel with locations as suggested. Here we show the Figure 3-5 with logarithmic colour scale to demonstrate our point.

---

## Author Comment (AC2)

This document contains the final response to comments from both reviewers on the original manuscript entitled "**The OCEAN ICE mooring compilation: a standardised, pan-Antarctic database of ocean hydrography and current time series**".

Reviewer's comments are in **blue**, and our point-by-point responses are in **black**.

**Reviewer #1:**

**General Comment.**

In this paper Zhou et al. present a large dataset compiling available moored observations of temperature, salinity and current velocities around Antarctica since the 1970s. The dataset is impressive, including close to 500 different datasets, covering a range of key ocean environments across all longitudes around Antarctica. Given the crucial role of Antarctic Ocean and ice processes on climate, extensive re-use of the dataset by the research is warranted and may enable substantial advances in the field. I am sure the community will be very grateful to the authors for their great efforts to put the dataset together. Therefore, I strongly endorse publication of the dataset and associated manuscript in ESSD. However, I have identified a series of potential issues and areas for improvement in the current version of the manuscript and dataset. It would be great if the authors could address my comments and suggestions, or at least respond to them, before I accept the article for final publication.

We thank reviewer's general comments on our manuscript. Our response to reviewer's comments below is in black.

**Specific Comments**

My main concern about the dataset is that it is not very clear how the user would know how much trust they could place on each individual instrument, dataset. Where should the user refer to get information about if and how instruments (e.g. conductivity cells) were calibrated, how was the data treated between acquisition and publication, and about general data quality and flags (and/or whether instrument failure or drift are flagged). I also understand this is a complex task, and it may not be realistic to recover detailed data quality metadata from all deployments, but it would be great if the authors could talk about this a bit more in the manuscript, acknowledge this (important) limitation and say that users should refer to the original datasets, but also give an indication on whether the original files contain some more information.

We agree that this information is not explicitly presented in the published dataset. It is indeed a technically challenging task to harmonise the level of quality control of all the datasets that were acquired with different level of processing. The quality of mooring timeseries has always been lack of consistency. But hopefully our dataset provides a standardised product to some degree. The source link provided in both the metadata

spreadsheet and each NetCDF file can lead users to the original source of the data or the data owner to better consult for the processing level of these data.

On a different note, it is great that the authors put together such a massive dataset. However, as I was reading the paper and having a look at the data, I have developed the feeling that the dataset can be a bit daunting, from the users' perspective, due to its richness and diversity. I would suggest that the authors do some effort to further digest the dataset to give a clearer overview of it and make it more accessible. Some suggestions I could come up with are:

- In the manuscript, present one or multiple figures and/or tables with general statistics on the dataset: e.g. periods and seasons covered, record lengths, regions sampled, depths sampled, how many records with TS, velocities, both, etc. One good thing to report would be some statistics on the instrument vertical location with respect to local bottom depth / surface, since you mention there is a "bias" toward deep measurements

We thank reviewer's thoughts on adding more general statistics to better overview the dataset. We included a new Figure 2 in the revised manuscript to illustrate the observing period in length and seasonal coverage, the fraction of each combination of variables measured by each mooring deployment, and averaged instruments depth (normalized) to demonstrate our point of 'bottom-biased' instrument installation. The exception for the shallower instruments is two under-ice turbulence clusters measurements in George XI ice shelf and Larsen C ice shelf, both deployed by BAS ice team. The instruments on these two moorings are deliberately mounted close to the ice base and within the ice to measure the ice temperature changes and ice-ocean boundary layer dynamics. The new Figure 2 in the revised manuscript is also attached below.

[Figure]

- In the datafiles. For each of the deployment files, you could add an overview in the metadata about the mooring location (not only latitude and longitude, but name of region), types of instruments and sampling depths and period. A link to the original dataset within files themselves could be useful. In general, richer deployment-specific metadata within each netcdf would be great. I am aware that modifying the whole dataset at this point could be a massive task, so please take this only as a suggestion, I leave to the authors' judgement whether this is a sensible thing to do.

We really appreciate reviewer's comments on including more metadata information. Although we believe that the naming of each mooring regions is always lack of consensus – in some cases the naming of the region can be very specific and in other cases it would have to be represented by the name of the broader area due to lack of naming system in mooring locations. We found that the longitude and latitude is the most consistent way of marking the mooring location.

- I wonder if the authors could share the code used to produce the figures for the preliminary analysis presented in the manuscript. That would give an example to users on how to bulk-access the data and generate some interesting insights from the ensemble of observations.

We believe that the code we used in producing figures and reading files are very basic MATLAB coding and should be easily replaced by other programming languages and can perhaps even be done better and more smartly by other more proficient MATLAB users. However, there is one fundamental piece of code we used for analysing tidal harmonics that needs a bit of instruction – the UTide toolbox. The UTide toolbox and its full instruction are available at https://uk.mathworks.com/matlabcentral/fileexchange/46523-utide-unified-tidal-analysis-and-prediction-functions.

**In line comments**

Line 98. Maybe provide an early indication of the size of the dataset: e.g. "This compilation includes 521 mooring time series [...]"

We thank reviewer's suggestion, but we feel that the mentioning of data quantity at the beginning of the following section makes a more coherent narrative overall. Considering it is not too far earlier in the text; we decided to keep the original way of mentioning the data quantity.

Line 109. Could you include some general description of the SOOS mooring dataset and how many more records you are including in your database.

We found that this point is always a tricky one to phrase, but in short, SOOS map does not provide any links to the actual dataset (in data archiving world, this is called landing page), in other words, there are no landing pages behind all the moorings appearing on SOOS mooring map other than some very brief information about the mooring (i.e., name, country, deployment status). In this database, we have included all mooring data that we can gathered from colleagues in the author list, excluding those are still in the water or being analysed/QC'ed by individuals and groups. Therefore, in terms of 'how many more', it would be 521 timeseries (OCEAN ICE) versus 0 timeseries (SOOSmap). We rephrased the text in the revised manuscript to make this point clearer.

Line 114. There is a typo here ",."

Corrected in the revised manuscript.

Figures 3-5. I felt the scatterplots may look better on log-colour scale, to highlight overall patterns rather than some particularly high values in some locations? Not sure… Also, I suggest annotating the maps with key locations mentioned in the text, e.g. on line 227 and others.

We thank reviewer's comments on colour scale. We agree that the logarithmic colour scale can highlight more detailed regional distribution, but these elevated small differences do not change the overall broad picture of the contrasting in non-tidal motion condition between cold and warm shelves, and the fact that tidal motion dominates the semi-diurnal, diurnal and fortnightly periodicities. Therefore, we decided to keep the linear colour scale in Figure 3-5, but we annotated all the first panel with locations as suggested. Here we show the Figure 3-5 with logarithmic colour scale to demonstrate our point.

[Figure]

**Figure 3**

[Figure]

**Figure 4**

[Figure]

**Figure 5**

Lines 201-214, Lines 222-232. I think these paragraphs would benefit from more support from citations.

We thank reviewer's comment in additional citations. Line 201-214 has been supported with additional citations. Line 222-232 is mainly our speculation, and we don't have direct supporting evidence from the existing literatures, in turn we propose that this speculation requires a more in-depth analysis and can be potentially a future research avenueS.

Line 212. "... mixing ..." Turbulence may be a more appropriate term here

Corrected in the original manuscript.

Figure 6. In the caption, can you add a bit more information about the different features presented in the TS plots here?

Additional information is added in the caption.

*Reviewer #2:*

**Summary**

This study presents the first standardized, pan-Antarctic compilation of moored hydrography and current time series, developed through a systematic analysis of historical mooring data from the marginal seas and contributions from international data centers, research institutes, and individual data providers. Spanning over five decades (1970s–2020s), the compilation enables detailed analysis of water mass transport and shelf connectivity across the Antarctic margin. The authors demonstrate the utility of the compilation through spectral analysis, removing dominant tidal signals via multi-linear regression. The detided records, though limited in duration, capture synoptic to seasonal variability, with regional patterns of kinetic energy offering insights for future study.

This dataset is a timely and valuable resource for research along the Antarctic margin, addressing an urgent need for sustained observations in the Southern Ocean. I appreciate the international effort by the group of observational scientists in assembling and standardizing this compilation. I have one major comment regarding Technical Quality, along with several minor comments detailed below. Overall, the manuscript is well written and provides a clear and well-structured contribution to the oceanographic community.

We thank reviewer's kind words on the manuscript, and we respond to the comments point-by-point below.

**Technical Quality**

While the compilation brings together an impressive range of historical mooring records, it remains unclear how the authors address uncertainty and error analysis across the dataset.

The manuscript notes that only minimal data cleaning was performed, with bad data identified by flags or unrealistic values replaced by NaNs, and no further interpolation or extrapolation applied. However, individual mooring datasets often include important quality control information, such as standard errors, instrument uncertainties, or confidence flags. A more detailed explanation of whether and how such uncertainty metrics were retained, harmonized, or reassessed would enhance the transparency of the compilation and support its appropriate scientific use, particularly given the analyses of detided variability presented in the study.

We appreciate reviewer's comments on the data quality control harmonisation. Similar feedback are raised by Reviewer #1. We acknowledge that the standardisation that we applied on the mooring timeseries is minimum. And this is mainly because that the raw dataset we received are generally lack of specific QC flags and often inconsistent in the data processing level. Mooring timeseries are unique type of dataset – data processing is often *ad hoc* and depending on individual/group. We therefore did not attempt to harmonise the QC flag but only provide minimum manipulation on the dataset and

meanwhile provide the original source link from which users can be advised to determine on user-defined processing needs.

– Figure 2: What causes the sharp drop in the black line at low frequencies in Figure 2a? Since this feature is not present in the detided spectra (Figure 2c), it may be related to tidal energy. A brief explanation would help clarify its origin. We agree that the dip in black line is odd. There is an error in the code, and we corrected it. We reproduced Figure 2; the reproduced black line has no dip anymore. See revised version here. (try to smooth the black line on linear/log space... it is good to have a thin black line anyway)

[Figure]

– Given the importance of the detided analysis, will the detided time series be made publicly available alongside the original data? We thank reviewer's comments on the de-tided timeseries. The implementation of UTide toolbox is straightforward, and there are a few user-defined options in the toolbox that can lead to slightly different results. We specify the user-defined parameters that we used in our study for reference in the revised manuscript, but we are cautious to publish the de-tided timeseries that we generated as we believe that users of these datasets should be entitled to generate their own version of de-tided timeseries. We acknowledge in the revised manuscript that the generated tidal harmonics and de-tided timeseries are served as first glance of the kinetic energy partition and there are a series user-defined options and parameters in

UTide toolbox that can be changed to tailor to users' needs, and the general guideline is put together by Codiga (2011).

It would also be helpful to clarify the sensitivity of the UTide method to record length, particularly in the case of shorter mooring records.

This is good point. Below we show the signal-noise ratio computed using UTide-estimated K1 magnitude divided by the UTide-estimate K1 uncertainty in function of the timespan of each mooring timeseries. There is a tendency in which the shorter record of mooring timeseries bears smaller signal-noise ratio Although the scatter plot tendency is skewed by the number of available moorings timeseries at different time span - most of the moorings have time span of 1-to-2 years which is the typical timescales of mooring turnaround. To account for the uneven data distribution over time span axis, the fitted line considers the data density using locally weighted linear regression method and the spread of the filled patch expands as less data become available at longer time span. The fitted line still shows an overall increase in signal-to-noise ratio with the observing time span, suggesting that the length of the record is a key factor in the robustness of estimated tidal signal using UTide toolbox.

[Figure]

- Lines 189 & 223: The authors define the seasonal band as spanning 80 days to 1.2 years. Would it be more appropriate to refer to this range as "seasonal-to-annual" to better reflect the upper bound?

We thank reviewer's suggestion on the choice of naming the timescales. It is rephrased throughout the revised manuscript.

- I reviewed the dataset and the provided spreadsheet, which includes useful information such as mooring file names, locations, time ranges, and DOI links to the original data sources. To improve usability, I suggest adding a summary document with more detailed metadata, including instrument type, depth ranges, variable definitions and units, quality control flags, and processing

history for each record. This additional information would significantly enhance the dataset's clarity and completeness. I also recommend sharing the analysis code used in the study to further support understanding and facilitate broader use of the dataset.

We thank reviewer's comments on the additional meta information. We append a new spreadsheet to the provided spreadsheet, which enclosed the instrument type, instrument depth, measuring variables and measuring period. See below the sample table from the additional spreadsheet.

| Filenames | Instrument | Velocity? | Temperature? | Salinity? | Measuring Depth | Start Date | End Date | Bottom Depth | Latitude |
|---|---|---|---|---|---|---|---|---|---|
| A.nc | Aanderaa_RCM5_01 | no | yes | yes | 175.1523493 | 724664.541 | 724693.9993 | 685 | -77.207 |
| A.nc | Aanderaa_RCM5_02 | yes | yes | yes | 353.6172251 | 724664.541 | 724997.791 | 685 | -77.207 |
| ADP1.nc | sbe37_01 | no | yes | yes | 786.067083 | 731646.8542 | 732004.0417 | 890 | -71.981 |
| ADP1.nc | sbe37_02 | no | yes | yes | 850.9389079 | 731646.8542 | 732004.0417 | 890 | -71.981 |
| ADP1.nc | sbe37_03 | no | yes | yes | 897.2067621 | 731646.8542 | 732004.0417 | 890 | -71.981 |
| ADP1.nc | Sontek_ADCP_C63 | yes | yes | no | 190.4785779 | 731646.875 | 732004.0417 | 890 | -71.981 |
| ADP2.nc | sbe37_01 | no | yes | yes | 511.7852621 | 732015.0208 | 732345.875 | 527 | -72.066 |
| ADP2.nc | sbe37_02 | no | yes | yes | 517.1765049 | 732015.0208 | 732345.875 | 527 | -72.066 |
| ADP2.nc | Sontek_ADCP_C180 | yes | yes | no | 499.799054 | 732015.0417 | 732221.1667 | 527 | -72.066 |
| AM01.nc | SBE37_1969 | no | yes | yes | 436.1832451 | 731231.7708 | 733780.2899 | 783 | -69.44203 |
| AM01.nc | SBE37_1970 | no | yes | yes | 574.8240797 | 731231.875 | 733414.5396 | 783 | -69.44203 |
| AM01.nc | SBE37_1971 | no | yes | yes | 734.1854552 | 731231.875 | 733355.9146 | 783 | -69.44203 |
| AM02.nc | SBE37_1174 | no | yes | yes | 762.3541379 | 730857.7917 | 733042.9987 | 790 | -69.7133 |
| AM02.nc | SBE37_1623 | no | yes | yes | 333.7359762 | 730857.7917 | 733042.9988 | 790 | -69.7133 |
| AM02.nc | SBE37_1624 | no | yes | yes | 555.5287787 | 730857.7917 | 733042.997 | 790 | -69.7133 |
| AM03.nc | SBE37_3883 | no | yes | yes | 1209.935669 | 732673 | 734494.9784 | 1254 | -70.561 |
| AM03.nc | SBE37_4054 | no | yes | yes | 860.5047205 | 732673 | 734494.9779 | 1254 | -70.561 |
| AM03.nc | SBE37_4055 | no | yes | yes | 652.7372946 | 732673 | 734494.9775 | 1254 | -70.561 |
| AM04.nc | SBE37_1972 | no | yes | yes | 535.109399 | 732688 | 734494.9792 | 931 | -69.9 |
| AM04.nc | SBE37_1973 | no | yes | yes | 674.6977769 | 732688 | 734494.9792 | 931 | -69.9 |
| AM04.nc | SBE37_1974 | no | yes | yes | 795.0922499 | 732688 | 734494.9792 | 931 | -69.9 |

The unit and definition of the variables are available in each individual NetCDF file which we decided not to repeatedly include in the spreadsheet. The updated spreadsheet is now available at SEANOE. We also included a summary diagram Figure 2 in revised manuscript to break down the measuring period, seasonal coverage and overall numbers of moorings for different type of measured variable combinations, and the abundance of instruments at different depth range. In response to reviewer's comments on sharing the analysing code used in this study, the scripts are mainly reading and extracting file information, which are not necessarily unique and can be done more cleanly and smartly. The only code that is not generated by us while important for producing the present results is the UTide toolbox that is publicly available at https://uk.mathworks.com/matlabcentral/fileexchange/46523-utide-unified-tidal-analysis-and-prediction-functions.

---

## Author Response (AR1)

This document contains the final response to comments from both reviewers on the original manuscript entitled "The OCEAN ICE mooring compilation: a standardised, pan-Antarctic database of ocean hydrography and current time series".

Reviewer's comments are in **blue**, and our point-by-point responses are in **black**. The changes made in the manuscript is annotated in red (line number in **bold**).

**Reviewer #1:**

**General Comment.**

In this paper Zhou et al. present a large dataset compiling available moored observations of temperature, salinity and current velocities around Antarctica since the 1970s. The dataset is impressive, including close to 500 different datasets, covering a range of key ocean environments across all longitudes around Antarctica. Given the crucial role of Antarctic Ocean and ice processes on climate, extensive re-use of the dataset by the research is warranted and may enable substantial advances in the field. I am sure the community will be very grateful to the authors for their great efforts to put the dataset together. Therefore, I strongly endorse publication of the dataset and associated manuscript in ESSD. However, I have identified a series of potential issues and areas for improvement in the current version of the manuscript and dataset. It would be great if the authors could address my comments and suggestions, or at least respond to them, before I accept the article for final publication.

We thank reviewer's general comments on our manuscript. Our response to reviewer's comments below is in black.

**Specific Comments**

My main concern about the dataset is that it is not very clear how the user would know how much trust they could place on each individual instrument, dataset. Where should the user refer to get information about if and how instruments (e.g. conductivity cells) were calibrated, how was the data treated between acquisition and publication, and about general data quality and flags (and/or whether instrument failure or drift are flagged). I also understand this is a complex task, and it may not be realistic to recover detailed data quality metadata from all deployments, but it would be great if the authors could talk about this a bit more in the manuscript, acknowledge this (important) limitation and say that users should refer to the original datasets, but also give an indication on whether the original files contain some more information.

We agree that this information is not explicitly presented in the published dataset. It is indeed a technically challenging task to harmonise the level of quality control of all the datasets that were acquired with different level of processing. Intercomparison of mooring timeseries has always suffered from a lack of consistency between mooring data types and processing. But hopefully our compilation provides a standardised

product to some degree. The source link provided in both the metadata spreadsheet and each NetCDF file can lead users to the original source of the data or the data owner to better assess the processing level of these data. We rephrased and added text to describe the data quality, acknowledge the concern raised by reviewer and entice future users to consider those concerns at **Line 162-166** in revised manuscript.

On a different note, it is great that the authors put together such a massive dataset. However, as I was reading the paper and having a look at the data, I have developed the feeling that the dataset can be a bit daunting, from the users' perspective, due to its richness and diversity. I would suggest that the authors do some effort to further digest the dataset to give a clearer overview of it and make it more accessible. Some suggestions I could come up with are:

• In the manuscript, present one or multiple figures and/or tables with general statistics on the dataset: e.g. periods and seasons covered, record lengths, regions sampled, depths sampled, how many records with TS, velocities, both, etc. One good thing to report would be some statistics on the instrument vertical location with respect to local bottom depth / surface, since you mention there is a "bias" toward deep measurements

We thank the reviewer's for their request to add more general statistics and propose an improved overview the dataset. We included a new Figure 2 in the revised manuscript to illustrate the observing period in length and seasonal coverage, the fraction of each combination of variables measured by each mooring deployment, and the averaged instruments depth (normalized) to demonstrate our point about the 'bottom-biased' nature of instrument installation in Antarctic shelf seas. The new Figure 2 in the revised manuscript is also attached below.

Figure 2 is added to the revised manuscript at **Line 150-155**. Text describing the mooring stats shown in Figure 2 and addressing reviewer's comments here is at **Line 143-149**.

• In the datafiles. For each of the deployment files, you could add an overview in the metadata about the mooring location (not only latitude and longitude, but name of region), types of instruments and sampling depths and period. A link to the original dataset within files themselves could be useful. In general, richer deployment-specific metadata within each netcdf would be great. I am aware that modifying the whole dataset at this point could be a massive task, so please take this only as a suggestion, I leave to the authors' judgement whether this is a sensible thing to do.

We really appreciate the reviewer's comments on including more metadata information. Although we believe that naming each mooring regions can be subective – in some cases the naming of the region can be very specific/localized and in other cases it would have to be represented by the name of the broader area due to lack of locally recognized feature/name. We found that the longitude and latitude is the most consistent way of marking the mooring location.

• I wonder if the authors could share the code used to produce the figures for the preliminary analysis presented in the manuscript. That would give an example to users on how to bulk-access the data and generate some interesting insights from the ensemble of observations.

We believe that the code we used in producing figures and reading files are very basic MATLAB coding and should be easily replaced by other programming languages and can perhaps even be done better by more proficient MATLAB users. However, there is one fundamental piece of code we used for analysing tidal harmonics that needs a bit of instruction – the UTide toolbox. The UTide toolbox and its full instruction are available at

https://uk.mathworks.com/matlabcentral/fileexchange/46523-utide-unified-tidal-analysis-and-prediction-functions.

**In line comments**

Line 98. Maybe provide an early indication of the size of the dataset: e.g. "This compilation includes 521 mooring time series [...]"

We thank reviewer's suggestion, but we feel that the mentioning of data quantity at the beginning of the following section makes a more coherent narrative overall. Considering

it is not too far earlier in the text; we decided to keep the original way of mentioning the data quantity.

Line 109. Could you include some general description of the SOOS mooring dataset and how many more records you are including in your database.

We found that this point is always a tricky one to phrase, but in short, SOOS map does not provide any links to the actual dataset (in data archiving world, this is called landing page), in other words, there are no landing pages behind all the moorings appearing on SOOS mooring map other than some very brief information about the mooring (i.e., name, country, deployment status). In our compilation, we have included all mooring data that we can gathered from colleagues in the author list, excluding those are still in the water or being analysed/QC'ed by individuals and groups. Therefore, in terms of 'how many more', it would be 521 timeseries (OCEAN ICE) versus 0 timeseries (SOOSmap). We rephrased the text in the revised manuscript to make this point clearer at Line 113-116.

Line 114. There is a typo here ",."

Corrected in the revised manuscript at Line 117.

Figures 3-5. I felt the scatterplots may look better on log-colour scale, to highlight overall patterns rather than some particularly high values in some locations? Not sure... Also, I suggest annotating the maps with key locations mentioned in the text, e.g. on line 227 and others.

We thank the reviewer for their comments on colour scale. We agree that the logarithmic colour scale can highlight more detailed regional distribution, but these elevated small differences do not change the overall broad picture of the contrasting in non-tidal motion condition between cold and warm shelves, and the fact that tidal motion dominates the semi-diurnal, diurnal and fortnightly periodicities. Therefore, we decided to keep the linear colour scale in Figure 3-5, but we annotated all the first panel with locations as suggested. Here, and to assuage the reviewer's curiosity, we show the Figure 3-5 with logarithmic colour scale to demonstrate our point. In the revised manuscripts we changed figures at Line 241, Line 273, Line 299.

Figure 3

Figure 4

\_

Lines 201-214, Lines 222-232. I think these paragraphs would benefit from more support from citations.

Line 201-214 are now supported with additional citations (Line 259-260). Line 222-232 is mainly our speculation, and we don't have direct supporting evidence from the existing literatures, in turn we propose that this speculation requires a more in-depth analysis and can be potentially a future research avenue, we added references relating to the different ice shelves here to support of our claim of cold/warm ice shelves (Line 283-293).

Line 212. "... mixing ..." Turbulence may be a more appropriate term here

Corrected in the original manuscript. (Line 268)

Figure 6. In the caption, can you add a bit more information about the different features presented in the TS plots here?

Additional information is added in the caption. (Line 320-326)

Reviewer #2:

**Summary**

This study presents the first standardized, pan-Antarctic compilation of moored hydrography and current time series, developed through a systematic analysis of historical mooring data from the marginal seas and contributions from international data centers, research institutes, and individual data providers. Spanning over five decades (1970s–2020s), the compilation enables detailed analysis of water mass transport and shelf connectivity across the Antarctic margin. The authors demonstrate the utility of the compilation through spectral analysis, removing dominant tidal signals via multi-linear regression. The detided records, though limited in duration, capture synoptic to seasonal variability, with regional patterns of kinetic energy offering insights for future study.

This dataset is a timely and valuable resource for research along the Antarctic margin, addressing an urgent need for sustained observations in the Southern Ocean. I appreciate the international effort by the group of observational scientists in assembling and standardizing this compilation. I have one major comment regarding Technical Quality, along with several minor comments detailed below. Overall, the manuscript is well written and provides a clear and well-structured contribution to the oceanographic community.

We thank reviewer's kind words on the manuscript, and we respond to the comments point-by-point below.

**Technical Quality**

While the compilation brings together an impressive range of historical mooring records, it remains unclear how the authors address uncertainty and error analysis across the dataset.

The manuscript notes that only minimal data cleaning was performed, with bad data identified by flags or unrealistic values replaced by NaNs, and no further interpolation or extrapolation applied. However, individual mooring datasets often include important quality control information, such as standard errors, instrument uncertainties, or confidence flags. A more detailed explanation of whether and how such uncertainty metrics were retained, harmonized, or reassessed would enhance the transparency of the compilation and support its appropriate scientific use, particularly given the analyses of detided variability presented in the study.

We appreciate the reviewer's comments on the data quality control harmonisation. Similar feedback is raised by Reviewer #1. We acknowledge that the standardisation that we applied on the mooring timeseries is minimal. And this is mainly because that the dataset we received generally lack specific QC flags and are often inconsistent in terms of data processing level. Mooring timeseries are unique types of dataset – data processing is often *ad hoc* and depending on individual/group. We therefore did not

attempt to harmonise the QC flag but only provide minimum manipulation on the dataset and meanwhile provide the original source links. We now further advise to determine on if further processing is needed depending on their needs. Additional texts addressing reviewer's concerns are at Line 162-166 in the revised manuscript.

**Minor comments**

Figure 2: What causes the sharp drop in the black line at low frequencies in Figure 2a? Since this feature is not present in the detided spectra (Figure 2c), it may be related to tidal energy. A brief explanation would help clarify its origin. We agree that the dip of the black line was odd. There was an error in the code, and we corrected it. We corrected black line in figure 2 has no dip anymore. See revised version below. The revised figure is used in the revised manuscript at Line 212.

Given the importance of the detided analysis, will the detided time series be made publicly available alongside the original data? We thank reviewer's comments on the de-tided timeseries. The implementation of UTide toolbox is straightforward, and there are a few user-defined options in the toolbox that can lead to slightly different results. We specify the user-defined parameters that we used in our study for reference in the revised manuscript, but we are reluctant to publish the de-tided timeseries that we generated as we believe that (1) users of these datasets should be entitled to generate their own version of de-tided timeseries and (2) this would unnecessarily increase the size of the dataset. We further acknowledge in the revised manuscript that the generated tidal

harmonics and de-tided timeseries are only used here for a first glance at the kinetic energy partition.

It would also be helpful to clarify the sensitivity of the UTide method to record length, particularly in the case of shorter mooring records.

This is a good point. Below we show the signal-to-noise ratio computed using UTide-estimated K1 magnitude divided by the UTide-estimate K1 uncertainty as a function of their timespan of each mooring timeseries. There is a natural/expected tendency for shorter records to bear smaller signal-to-noise ratio. Although the scatter plot tendency is skewed by the number of available moorings timeseries at different time span - most of the moorings have time span of 1-to-2 years which is the typical timescales of mooring turnaround. To account for the uneven data distribution over the time span axis, the fitted line considers the data density using a locally weighted linear regression method and the spread of the filled patch expands as less data become available at longer time span. The fitted line still shows an overall increase in signal-to-noise ratio with the observing time span, suggesting that the length of the record is a key factor in the robustness of estimated tidal signal using UTide toolbox.

- Lines 189 & 223: The authors define the seasonal band as spanning 80 days to 1.2 years. Would it be more appropriate to refer to this range as "seasonal-toannual" to better reflect the upper bound?
  - We thank the reviewer's suggestion on the choice of naming the timescales. It is now rephrased throughout the revised manuscript.
- I reviewed the dataset and the provided spreadsheet, which includes useful
  information such as mooring file names, locations, time ranges, and DOI links to
  the original data sources. To improve usability, I suggest adding a summary
  document with more detailed metadata, including instrument type, depth
  ranges, variable definitions and units, quality control flags, and processing

history for each record. This additional information would significantly enhance the dataset's clarity and completeness. I also recommend sharing the analysis code used in the study to further support understanding and facilitate broader use of the dataset.

We have now appended a new spreadsheet to the provided spreadsheet, which contains the instrument type, instrument depth, observed variables and observational period. See below the sample table from the additional spreadsheet. The updated spreadsheet information is described in the Data availability session at Line 394-400.

| Filenames | Instrument       | Velocity? | Temperature? | Salinity? | Measuring Depth | Start Date  | End Date    | Bottom
Depth | Latitude  |
|-----------|------------------|-----------|--------------|-----------|-----------------|-------------|-------------|-----------------|-----------|
| A.nc      | Aanderaa_RCM5_01 | no        | yes          | yes       | 175.1523493     | 724664.541  | 724693.9993 | 685             | -77.207   |
| A.nc      | Aanderaa_RCM5_02 | yes       | yes          | yes       | 353.6172251     | 724664.541  | 724997.791  | 685             | -77.207   |
| ADP1.nc   | sbe37_01         | no        | yes          | yes       | 786.067083      | 731646.8542 | 732004.0417 | 890             | -71.981   |
| ADP1.nc   | sbe37_02         | no        | yes          | yes       | 850.9389079     | 731646.8542 | 732004.0417 | 890             | -71.981   |
| ADP1.nc   | sbe37_03         | no        | yes          | yes       | 897.2067621     | 731646.8542 | 732004.0417 | 890             | -71.981   |
| ADP1.nc   | Sontek_ADCP_C63  | yes       | yes          | no        | 190.4785779     | 731646.875  | 732004.0417 | 890             | -71.981   |
| ADP2.nc   | sbe37_01         | no        | yes          | yes       | 511.7852621     | 732015.0208 | 732345.875  | 527             | -72.066   |
| ADP2.nc   | sbe37_02         | no        | yes          | yes       | 517.1765049     | 732015.0208 | 732345.875  | 527             | -72.066   |
| ADP2.nc   | Sontek_ADCP_C180 | yes       | yes          | no        | 499.799054      | 732015.0417 | 732221.1667 | 527             | -72.066   |
| AM01.nc   | SBE37_1969       | no        | yes          | yes       | 436.1832451     | 731231.7708 | 733780.2899 | 783             | -69.44203 |
| AM01.nc   | SBE37_1970       | no        | yes          | yes       | 574.8240797     | 731231.875  | 733414.5396 | 783             | -69.44203 |
| AM01.nc   | SBE37_1971       | no        | yes          | yes       | 734.1854552     | 731231.875  | 733355.9146 | 783             | -69.44203 |
| AM02.nc   | SBE37_1174       | no        | yes          | yes       | 762.3541379     | 730857.7917 | 733042.9987 | 790             | -69.7133  |
| AM02.nc   | SBE37_1623       | no        | yes          | yes       | 333.7359762     | 730857.7917 | 733042.9988 | 790             | -69.7133  |
| AM02.nc   | SBE37_1624       | no        | yes          | yes       | 555.5287787     | 730857.7917 | 733042.997  | 790             | -69.7133  |
| AM03.nc   | SBE37_3883       | no        | yes          | yes       | 1209.935669     | 732673      | 734494.9784 | 1254            | -70.561   |
| AM03.nc   | SBE37_4054       | no        | yes          | yes       | 860.5047205     | 732673      | 734494.9779 | 1254            | -70.561   |
| AM03.nc   | SBE37_4055       | no        | yes          | yes       | 652.7372946     | 732673      | 734494.9775 | 1254            | -70.561   |
| AM04.nc   | SBE37_1972       | no        | yes          | yes       | 535.109399      | 732688      | 734494.9792 | 931             | -69.9     |
| AM04.nc   | SBE37_1973       | no        | yes          | yes       | 674.6977769     | 732688      | 734494.9792 | 931             | -69.9     |
| AM04.nc   | SBE37_1974       | no        | yes          | yes       | 795.0922499     | 732688      | 734494.9792 | 931             | -69.9     |

The unit and definition of the variables are available in each individual NetCDF file which we decided not to repeatedly include in the spreadsheet. The updated spreadsheet is now available at SEANOE. We also included a summary diagram Figure 2 in the revised manuscript to break down the measuring period, seasonal coverage and overall numbers of moorings for different type of measured variable combinations, and the abundance of instruments at different depth range. Concerning the reviewer's comments on sharing the analysing code used in this study, the scripts are mainly reading and extracting file information, which are not necessarily unique and can be easily replicated. The only code that is not generated by us and is more involved while important for producing the present results is the UTide toolbox that is publicly available at

 $\frac{https://uk.mathworks.com/matlabcentral/file exchange/46523-utide-unified-tidal-analysis-and-prediction-functions.}{}$